# Ovicidal, larvicidal and pupicidal efficacy of silver nanoparticles synthesized by *Bacillus marisflavi* against the chosen mosquito species

**Thelma J.** ◉¤\*, **Balasubramanian C.**

Department of Zoology, Thiagarajar College, Madurai, Tamil Nadu, India

¤ Current address: Department of Zoology, Fatima College, Madurai, Tamil Nadu, India
\* thelmaphd2016@gmail.com

**Data Availability Statement:** All relevant data are within the manuscript and its Supporting information files.

## Abstract

Microbial synthesis of silver nanoparticles is more advantageous and is eco-friendly to combat the various vectors that cause diseases in humans. Hence, in the present study a *Bacillus* strain is isolated from marine habitat and is evaluated for its ability to synthesize silver nanoparticles (AgNPs) and its efficacy evaluated against the immature stages of selected mosquito species. The effective candidate was confirmed to be *Bacillus marisflavi* after 16S rRNA sequencing. The synthesis of AgNPs was confirmed by UV-Vis spectro-photometer. Atomic Force Microscopic (AFM) analysis showed spherical nanoparticles. Size analysis using Scanning Electron Microscope (SEM) showed particles of nano size averaging 78.77 nm. The diameter of the particles analyzed by Dynamic Light Scattering (DLS) showed 101.6 nm with a poly-dispersive index of 0.3. Finally the elemental nature of the nanoparticles was identified by Fourier-transform infrared spectroscopy (FTIR). $LC_{50}$ and $LC_{90}$ values for the ovicidal, larvicidal and pupicidal efficacy of the AgNPs against the egg, larvae and pupae of *Aedes aegypti*, *Culex quinquefasciatus* and *Anopheles stephensi* respectively were evaluated. The present study revealed that the nanoparticles have an excellent toxic effect against the disease transmitting vector mosquitoes. Hence, the rapid synthesis of AgNPs would be an appropriate eco-friendly tool for biocontrol of vector mosquitoes.

## Introduction

Mosquitoes are small midge like flies that transmit fatal diseases to humans. Malaria, filariasis, encephalitis, dengue and chikungunya are the leading mosquito borne diseases in India. About 110 genera and 3000 species of mosquitoes have been recorded worldwide [1], but three species namely *Ae. aegypti*, *Cx quinquefasciatus* and *An. stephensi* are accountable for spreading vector-borne diseases in humans all over the world. These vector-borne diseases are one of the major concerns of health problems in developing countries like India [2]. The control of

**Funding:** The author(s) received no specific funding for this work.

**Competing interests:** The authors have declared that no competing interests exist.

mosquito species can be made possible by the use of physical and chemical methods; but these methods aggravate the resistance in mosquitoes. To overcome these problems of insecticide resistance by mosquitoes, Gram-positive entomopathogenic bacterial toxin has been considered as a reliable source in the control of insect pests both in agriculture and in vectors of human diseases. In this view, *B. thuringiensis (Bt)* has been extensively used to control the pest insects. The sporulated spray of *Bt*, *Bacillus sphaericus (Bsp)* and *B.thuringiensis subsp. israelensis (Bti)* have been identified as potential biocontrol agents of human vector mosquitoes [3]. Some studies have highlighted that mosquito species have developed resistance against the toxins of entomopathogenic bacteria [4]. Hence, the nanoparticles synthesized by these bacterial species have proven to overcome the problem of insecticide resistance. Nanoparticles are particulate matters that measure 100 nm in a nanometer scale. The area of nanoparticle research is getting more fascinating and is gaining more importance because of their catalytic activity, optical property, electro-magnetic property and antimicrobial activity [5–7]. Nanoparticles can be synthesized by plenty of approaches including physical, chemical and biological methods [8, 9]. The demerits of using physical and chemical methods are that they pose a biomagnification. There is an increased demand for the nanoparticles synthesized by microbes since they are eco-friendly and less expensive when compared to physical and chemical methods [10, 11]. Microbes are capable of synthesizing AgNPs extracellularly and intracellularly. Intracellular synthesis of AgNPs needs expensive down streaming process and hence extracellular synthesis of AgNPs serves as a hopeful and economical method [12–15]. Above all, the sources of bacteria have been inexpensive and efficient to control immature stages of mosquitoes to minimize the incidence of vector-borne diseases from environment. Therefore, the development of bacterial larvicides for mosquito control appears to be extremely significant. Entomopathogenic bacteria belonging to the genus *Bacillus* have been studied extensively but studies on the effect of nanoparticles synthesized extracellularly by the members of the genus *Bacillus* appears to be very limited. The nanoparticles secreted in a biogenic mode contain large amount of proteins that are responsible for the reduction of metal ions with controlled morphology [16]. Therefore, the present study is intended to explore the nanoparticle synthesizing ability of marine *Bacillus* species and its efficacy is evaluated against various stages such as egg, larva and pupa of *Ae. aegypti*, *Cx. quinquefasciatus* and *An. stephensi* (L.) respectively under laboratory condition.

## Materials and methods

### Sample collection

*Bacillus* species used in this study was isolated from marine water samples collected from Marina Beach (Latitude: 13˚03'15.05"N; Longitude: 80˚ 17' 1.25" E), Chennai, Tamil Nadu, India. The sea water samples collected from 0.3m below the water surface using polyethylene containers were stored in ice and transported to laboratory. The samples were stored in refrigerator at 4˚C for further work.

### Isolation of marine *Bacilli*

One ml of sea water sample was suspended in 9 ml of sterile distilled water and serially diluted. For isolating the pure culture of *Bacillus*, the dilutions were incubated at 30˚C for 4 hours and heat shocked at 80˚C for 3 to 5 minutes in a water bath to destroy all vegetative microbial cells. 100μl of sample from each dilution was taken and spread on Zobell marine agar (HiMedia, India). The plates were inverted and incubated at 30˚C for 24 hrs.

## Staining and identification

Pure colonies were picked up from the agar plate by using inoculation loop and were subjected to Gram's staining. The isolated colonies were identified on the basis of morphological features such as the half-white, uneven edges and richly grown colonies [17].

## Species identification

In order to characterize the organism at species level, the strains were subjected to genetic characterization by 16S rRNA gene sequencing. The Chromosomal DNA was isolated by using standard protocols. The isolated DNA was amplified by PCR with specific primers for the conserved regions of 16S rRNA following which the DNA was sequenced. The sequences were compared with sequences deposited in the 16S rRNA database.

## Phylogenetic analysis

The sequences were submitted in GenBank and BLAST was performed to identify the reference dataset for phylogenetic tree construction. Phylogenetic analysis was done using MEGA 7 software and the phylogenetic tree was constructed using the sequences generated from this study along with 18 reference sequences retrieved from GenBank, NCBI. Tree construction using Maximum Likelihood method (ML) with bootstrap value of 1000 times replication and gaps were considered as missing data [18].

## Extracellular biosynthesis of AgNPs using culture supernatant

Extracellular synthesis of AgNPs was carried out as described by [19] with slight modifications. The isolated colonies were sub-cultured in Zobell marine broth and incubated for 24 hrs at 37°C. The broth was centrifuged (Remi, Laboratory Centrifuge, Mumbai, India) at 8000 rpm for 10 min to collect the culture supernatant. 1mM (millimolar) silver nitrate (Laboratory Reagent, Reachem Laboratory Chemicals Private Ltd, Chennai, Tamil Nadu, India) solution was prepared in double distilled water. 200 ml of aqueous solution of 1mM silver nitrate was treated with 100 ml of culture supernatant in a 500 ml Erlenmeyer flask. The whole sample was kept in the orbit incubator shaker (Neolab, Neolab instruments, Mumbai, India) at 150 rpm and maintained in dark condition for 72 hrs at room temperature. The reduction of silver nitrate was monitored by visible color change of the solution.

## Characterization of AgNPs

The morphology of the AgNPs was determined by Scanning Electron Microscope (SEM) (Make: CAREL ZEISS, Model: EVO 18) and Atomic Force Microscopy (AFM) (JEOL 4210). The chemical composition and the size of the AgNPs were characterized by Fourier-transform infrared spectroscopy (FTIR) (Perkin Elmer-Spectrum Two FT-IR spectrometer with OPUS software in the range 4000–400 cm$^{-1}$, at a resolution of 4cm$^{-1}$) and Dynamic Light Scattering (DLS)(Make: Micromeritics; Model: Nano Plus).

## Evaluation of ovicidal activity

The ovicidal activity of the AgNPs was evaluated by modified method of Reegan et al. [20]. Eggs of *An. stephensi*, *Cx. quinquefasciatus* and *Ae. Aegypti* were procured from ICMR, Madurai, Tamil Nadu, India. The AgNPs synthesized by *Bacillus* species were prepared in various concentrations ranging from 5ppm to 80 ppm. The AgNPs synthesized by *Bacillus thuringiensis (Bt)* (MTCC strain code: 9025) species were used as positive control for comparison. For each concentration, twenty five freshly laid eggs of mosquito species were exposed, and each

concentration was replicated four times. The replications were maintained separately and were covered with a mosquito net. After treatment, hatching assessments were done 120 h post-treatment by the following formula.

$$\% \text{ of ovicidal activity} = \frac{\text{Number of unhatched eggs} \times 100}{\text{Total number of eggs introduced}}$$

## Evaluation of the larvicidal activity

Larvicidal bioassay of the formulation was performed according to the method described by World Health Organization [21] for bacterial larvicides. The early 3rd and 4th instar larvae of *An. stephensi*, *Cx. quinquefasciatus* and *Ae. aegypti* were used for the study. The larvae were obtained from ICMR, Madurai, Tamil Nadu, India. For bioassay, 25 larvae for each concentration were transferred into 250 mL glass beaker (Borosil®) containing 5ppm to 80 ppm concentration of microbial AgNPs. The AgNPs synthesized by *Bacillus thuringiensis* (MTCC strain code: 9025) were used as positive control for comparison. Each concentration was replicated four times. The replications were maintained separately and were covered with a mosquito net. The larvae were provided a mixture of dog biscuit and yeast powder in a 3:1 ratio as described by Kamaraj et al. [22]. The experiments were carried out at 26°C ± 2°C. Mortality of larvae was monitored at 24 hours. Larvae were considered dead when they did not move when disturbed using a glass rod. Percent mortality was calculated using formula (1) and corrections for mortality when necessary were done using Abbot's [23] formula (2).

Percentage of mortality:

$$\frac{\text{No of dead larvae}}{\text{No of larvae introduced}} \text{ X } 100 \tag{1}$$

Corrected percentage of mortality:

$$\frac{1 - n \text{ in T after treatment X } 100}{n \text{ in C after treatment}} \text{ X } 100 \tag{2}$$

Where, n is the number of larvae, T—treated and C is the control.

## Evaluation of pupicidal activity

To evaluate the pupicidal activities of the AgNPs, the procedure described by Kovendan et al. was followed [24]. Twenty-five individuals of freshly emerged pupae were kept in a 250 mL glass beaker containing dechlorinated water to which AgNPs synthesized by the *Bacillus* species was added in different concentrations ranging from 5ppm to 80 ppm. The AgNPs synthesized by *Bacillus thuringiensis* (MTCC strain code: 9025) were used as positive control for comparison. For each concentration, four replications were maintained. All the experiments were carried out at 26°C ± 2°C. Mortality of larvae was monitored at 24 hours. Pupae were considered dead when they did not move when disturbed using a glass rod. Percent mortality was calculated using the formula (1) and corrections for mortality when necessary were done using Abbot's formula (2).

Percentage of mortality:

$$\frac{\text{No of dead pupae}}{\text{No of pupae introduced}} \text{ X } 100 \tag{1}$$

Corrected percentage of mortality:

$$\frac{1 - n \text{ in } \text{T after treatment X 100}}{n \text{ in } \text{C after treatment}} \text{ X } 100 \tag{2}$$

Where, n is the number of pupae, T—treated and C is the control.

## Statistical analysis

The average percentage mortality, $LC_{50}$, $LC_{90}$ and Chi-square values were calculated using GraphPad Prism 8 software. Results with $P < 0.05$ were considered statistically significant.

## Results and discussion

### Bacterial identification & phylogenetic analysis

The sea water samples were serially diluted and the colonies that had the morphology of *Bacilli* were isolated by specific procedures and characterized. Based on biochemical characterization (Fig 1) and 16S rRNA sequencing (Fig 2), the organism was identified to be *B.marisflavi*. The sequence has been deposited in GenBank under accession No. MN368726.

The evolutionary history was inferred using the Maximum Likelihood method based on the Tamura-Nei model [25]. The tree with the highest log likelihood (-2237.17) is shown in Fig 3. The percentage of trees in which the associated taxa were clustered together is shown next to the branches. Initial tree(s) for the heuristic search were obtained automatically by applying Neighbor-Join and BioNJ algorithms to a matrix of pairwise distances, estimated using the Maximum Composite Likelihood (MCL) approach, and then selecting the topology with superior log likelihood value. The tree is drawn to scale, with branch lengths measured in the number of substitutions per site. The analysis involved 20 nucleotide sequences. All positions with less than 95% site coverage were eliminated; i.e. fewer than 5% alignment gaps, missing data, and ambiguous bases were allowed at any position. There were a total of 1396 positions in the final dataset. Evolutionary analyses were conducted in MEGA7 [26].

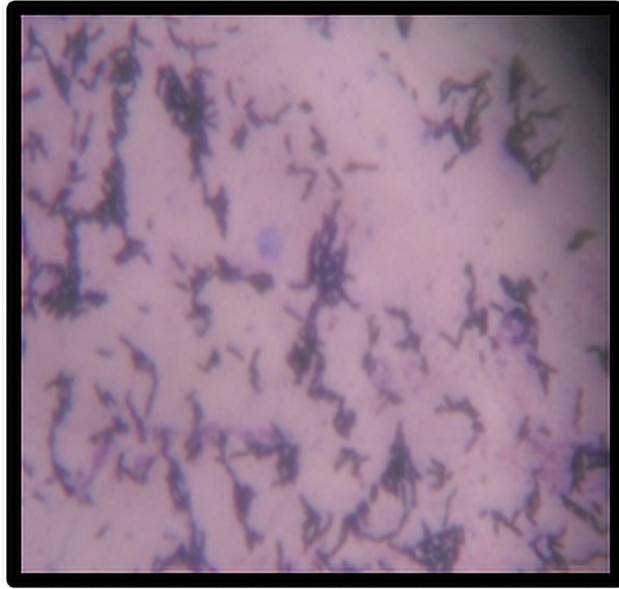

**Fig 1. Gram stain of the selected *Bacillus* strain showing gram-positive rods.**

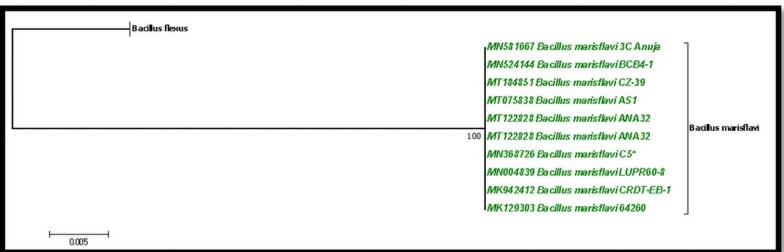

```
>Bacillus marisflavi
AGCGGATCGATGGGAGCTTGCTCCCTGAGATCAGCGGCGGACGGGTGAGTAACACGTGGGTAACCTGCCTGTAAGACTGGGATAACTCCGGGAAACCGGG
GCTAATACCGGATAACACCTACCCCCGCATGGGGGAAGGTTGAAAGGTGGCTTCGGCTATCACTTACAGATGGACCCGCGGCGCATTAGCTAGTTGGTGAG
GTAATGGCTCACCAAGGCGACGATGCGTAGCCGACCTGAGAGGGTGATCGGCCACACTGGGACTGAGACACGGCCCAGACTCCTACGGGAGGCAGCAGT
AGGGAATCTTCCGCAATGGACGAAAGTCTGACGGAGCAACGCCGCGTGAGTGAAGAAGGTTTTCGGATCGTAAAACTCTGTTGTTAGGGAAGAACAAGT
GCCGTTCGAATAGGGCGGCGGCCTTGACGGTACCTAACCAGAAAGCCACGGCTAACTACGTGCCAGCAGCCGCGGTAATACGTAGGTGGCAAGCGTTGTCC
GGAATTATTGGGCGTAAAGCGCGCGCAGGTGGTTTCTTAAGTCTGATGTGAAAGCCCACGGCTCAACCGTGGAGGGTCATTGGAAACTGGGGAACTTGAG
TGCAGAAGAGGAAAGTGGAATTCCAAGTGTAGCGGTGAAATGCGTAGATATTTGGAGGAACACCAGTGGCGAAGGCGACTTTCTGGTCTGTAACTGACAC
TGAGGCGCGAAAGCGTGGGGAGCAAACAGGATTAGATACCCTGGTAGTCCACGCCGTAAACGATGAGTGCTAAGTGTTAGAGGGTTTCCGCCCTTTAGTGC
TGCAGCTAACGCATTAAGCACTCCGCCTGGGGAGTACGGTCGCAAGACTGAAACTCAAAGGAATTGACGGGGGCCCGCACAAGCGGTGGAGCATGTGGT
TTAATTCGAAGCAACGCGAAGAACCTTACCAGGTCTTGACATCCTCTGACAACCCTAGAGATAGGGCTTTCCCCTTCGGGGGACAGAGTGACAGGTGGTG
CATGGTTGTCGTCAGCTCGTGTCGTGAGATGTTGGGTTAAGTCCCGCAACGAGCGCAACCCTTGATCTTAGTTGCCAGCATTCAGTTGGGCACTCTAAGAT
GACTGCCGGTGACAAACCGGAGGAAGGTGGGGATGACGTCAAATCATCATGCCCCTTATGACCTGGGCTACACACGTGCTACAATGACGGTACAAAGGG
CTGCAAGACCGCGAGGTTTAGCCAATCCCATAAAACCGTTCTCAGTTCGGATTGTAGGCTGCAACTCGCCTACATGAAGCTGGAATCGCTAGTAATCGCG
GATCAGCATGCCGCGGTGAATACGTTCCCGGGCCTTGTACACACCGCCCGTCACACCACGAGAGTTTGTAACACCCGAAGTCGGTGAGGTAAC
```

**Fig 2. 16srRNA sequence of *Bacillus marisflavi*.**

## Synthesis and characterization of AgNPs

The selected strain was tested for its ability to synthesize AgNPs. Based on the visible color change observed in the AgNO₃ solution with culture supernatant (Fig 4), it was evident that the strain was capable of synthesizing AgNPs. Studies by Sheny et al; Vilas et al. and Gaal et al. [27–30] have also reported the color change during the synthesis of AgNPs. This was confirmed by UV spectrophotometric analysis in which the peak was between 420nm– 430nm

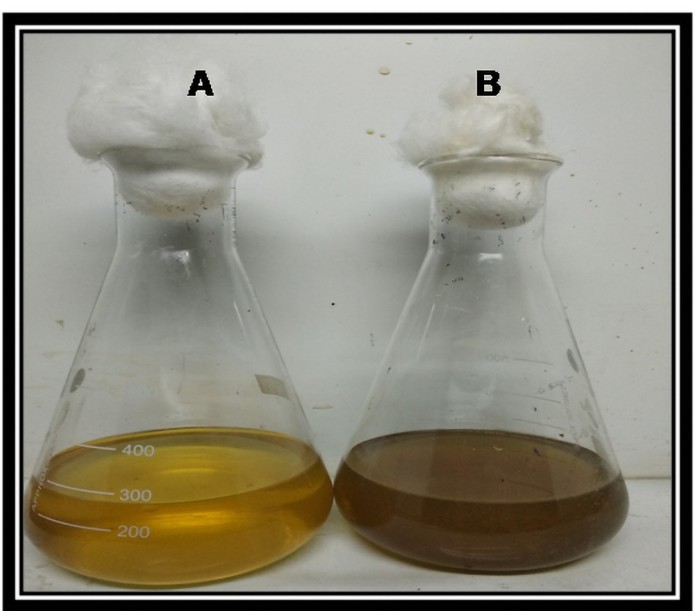

**Fig 3. Molecular phylogenetic analysis by maximum likelihood method.**

**Fig 4. Synthesis of AgNPs by *B.marisflavi*.** A: Zobell marine broth (control) + 1mM Silver nitrate (no color change); B: Culture supernatant of *B. marisflavi* + 1mM Silver nitrate (visible color change from yellow to brown).

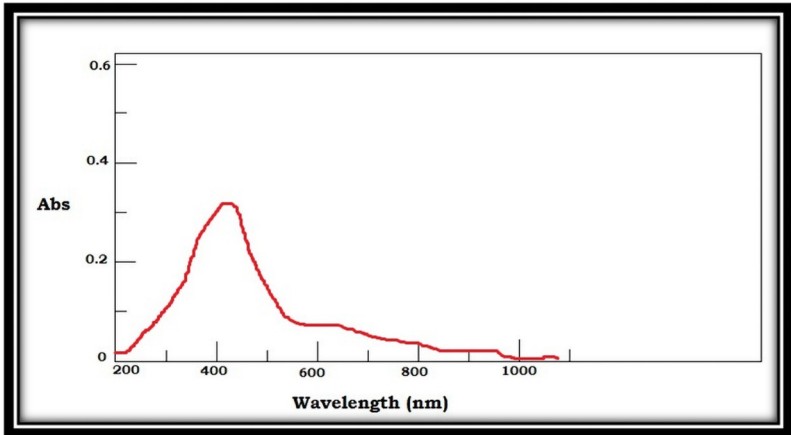

**Fig 5. UV-VIS spectrum showing maximum absorption between 420-430nm.**

(Fig 5). The Plasmon resonance of the electrons on the surface of silver nanoparticles is the reason for the formation of UV spectrum at this range [31]. Other bacterial species such as *E. coli* [13] and *Klebsiella pneumonia* [14] have been reported to synthesize AgNPs at this range. Culture supernatant of *B.marisflavi* isolated from the agricultural waste samples showed peak between 420nm to 430nm [32]. Further, the shape and surface topography of AgNPs were analyzed using AFM. Spherical nanoparticles were found to be present (Fig 6), which is in

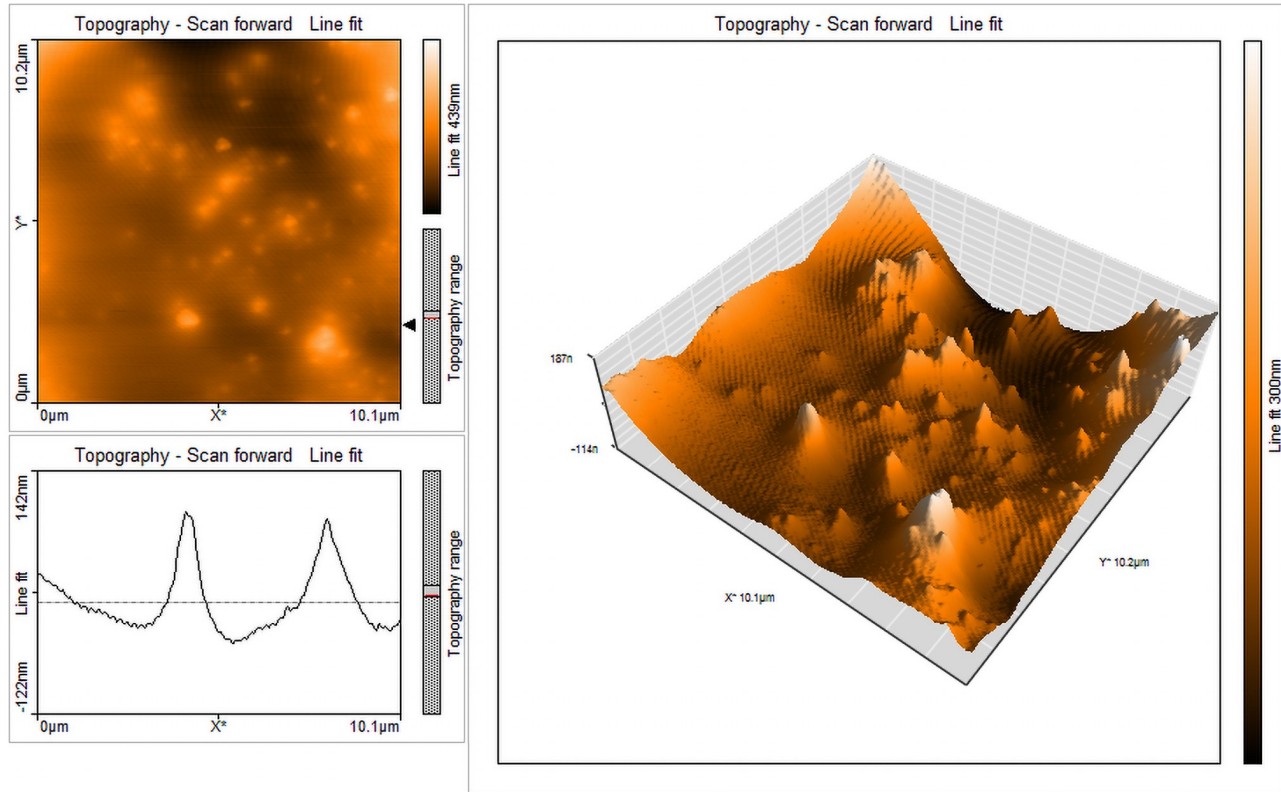

**Fig 6. AFM results showing spherical nanoparticles.**

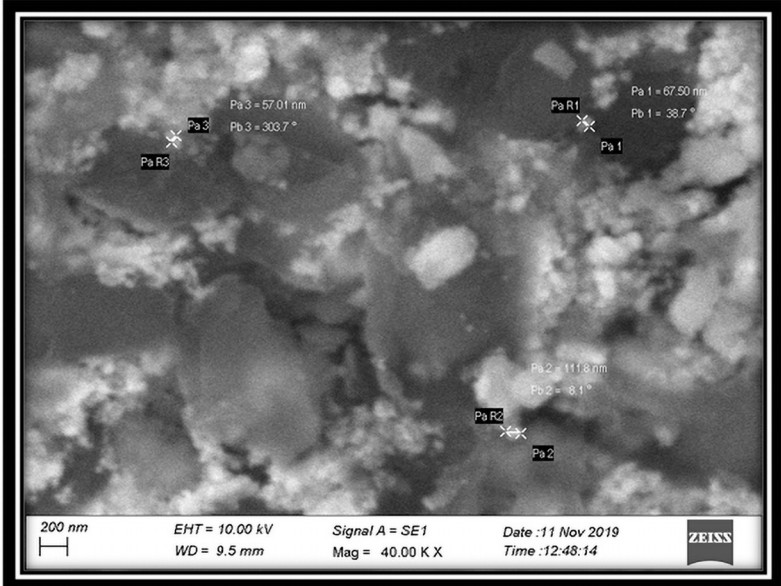

**Fig 7. SEM image of silver nanoparticles of nano size averaging 78.77 nm.**

concordance with the AgNPs synthesized by *Sesbania grandiflora* [33], *Pterocarpus santalinus* leaf extracts [34], *Bacillus flexus* [35] and *Bacillus* species GP23 [36]. SEM analysis showed AgNPs of nano size averaging 78.77 nm (Fig 7). A study by Srikar et al. [37] exposed that synthesis of nanoparticles means production of particles with sizes less than 100 nm. Similar results were also observed in AgNPs synthesized by *Streptococcus thermophilus* with size of 28–122nm [38]. The DLS and SEM analysis of the AgNPs synthesized by chemical reduction methods showed the presence of AgNPs with size 79.22 nm and 80.32nm respectively [39]. The DLS results of the present study showed the diameter of AgNPs to be 101.6 nm with a polydispersity index of 0.3 (Fig 8). Similar results were observed in AgNPs synthesized by

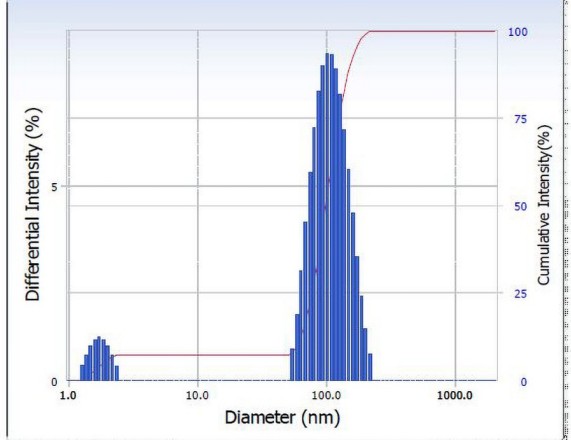
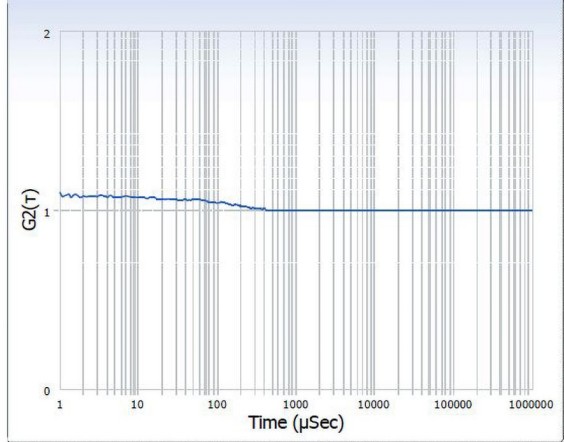

**Fig 8. DLS results showing the diameter and polydispersity index of the silver nanoparticles.**

**Table 1. Functional groups corresponding to different absorption peaks obtained for the AgNPs.**

| S. No | Absorption peaks | Functional groups | Compound Class |
|---|---|---|---|
| 1 | 3424.69 cm$^{-1}$ | N-H stretching | primary amine |
| 2 | 2924.09 cm$^{-1}$ | C-H stretching | alkane |
| 3 | 2855.09 cm$^{-1}$ | C-H stretching | alkane |
| 4 | 2354.31 cm$^{-1}$ | S-H stretching | Thiol |
| 5 | 2051.78 cm$^{-1}$ | N=C=S | Isothiocyanate |
| 6 | 1726.26 cm$^{-1}$ | C=O | aldehyde |
| 7 | 1636.76 cm$^{-1}$ | C=C stretching | conjugated alkene |
| 8 | 1543.76 cm$^{-1}$ | N-O stretching | nitro compound |
| 9 | 1430.43 cm$^{-1}$ | O-H bending | carboxylic acid |
| 10 | 1383.03 cm$^{-1}$ | S=O stretching | sulfate |
| 11 | 1319.39 cm$^{-1}$ | C-O stretching | aromatic ester |
| 12 | 1280.84 cm$^{-1}$ | C-N stretching | aromatic amine |
| 13 | 1234.46 cm$^{-1}$ | C-O stretching | alkyl aryl ether |
| 14 | 1023.13 cm$^{-1}$ | S=O stretching | Sulfoxide |
| 15 | 660.49 cm$^{-1}$ | C-Br stretching | halo compound |
| 16 | 598.52 cm$^{-1}$ | C-I stretching | halo compound |
| 17 | 553.58 cm$^{-1}$ | C-I stretching | halo compound |

extracts of *Oscillatoria sp* [40] with a diameter of 558.1 nm and polydispersity index of 0.580. Nanoparticles with polydispersity index of 0.3 are mostly used in drug delivery system [41–43]. The elemental analysis of the AgNPs done by FTIR analysis showed the presence of compounds as listed in Table 1. The absorption spectra are indicated in Fig 9. The peak formed at 3424.69 cm-1 suggests the presence of N-H stretching. The peak at 2924.09 cm$^{-1}$ and 2855.09 cm$^{-1}$ indicates the presence of C-H stretching. The presence of S-H stretching was confirmed by the peak formation at 2354.31 cm$^{-1}$. The presence of N=C=S was indicated by the peak formation at 2051.78 cm$^{-1}$. The peak at 1726.26 cm$^{-1}$ is C=O and peak at 1636.76 cm$^{-1}$ is C=C

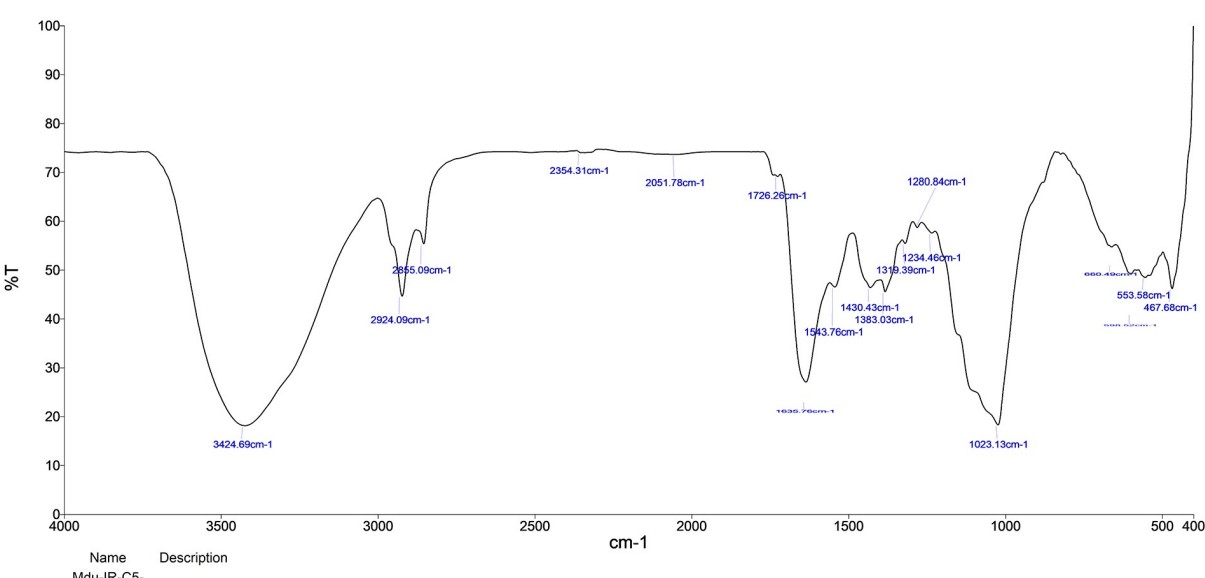

**Fig 9. FTIR results showing the presence of different chemical groups in the silver nanoparticles.**

stretching. The peak at 1543.76 cm$^{-1}$ indicated the presence of N-O stretching and the peak at 1430.43 cm$^{-1}$ is O-H bending. The peak formed at 1383.03 cm$^{-1}$ and 1023.13 cm$^{-1}$ confirms the presence of S=O stretching. C-O is confirmed by the presence of peaks formed at 1319.39 cm$^{-1}$ and 1234.46 cm$^{-1}$. The peak 1280.84 cm$^{-1}$ is C-N and the peak 660.49 cm$^{-1}$ is C-Br stretching. The peak formed at 598.52 cm$^{-1}$ and 553.58 cm$^{-1}$ suggests the presence of C-I stretching. This clearly indicates that the AgNPs are doped with significant phytocomponents.

## Evaluation of ovicidal, larvicidal and pupicidal efficacy of AgNPs

The synthesized AgNPs were evaluated for the ovicidal, larvicidal and pupicidal activity against the selected mosquito species namely, *Ae. aegypti*, *Cx. quinquefasciatus* and *An. stephensi*. In the present work, the AgNPs exhibited significant ovicidal activity. The percentage of ovicidal activity revealed that *An. stephensi* was more susceptible to the activity of AgNPs followed by *Cx. Quinquefasciatus* and *Ae. Aegypti* (Table 2, Fig 10). The LC$_{50}$ and LC$_{90}$ values for the ovicidal activity were 13.96ppm and 63.31ppm, 24.54ppm and 69.93ppm, 29.14ppm and 65.84ppm for *Ae. aegypti*, *Cx. Quinquefasciatus* and *An. stephensi*, respectively (Table 3). The ethanolic leaf extracts of *Duranta erecta*, *Tridax procumbens* and *Pennisetum purpureum* showed ovicidal activity against *An. gambiae*. *D. erecta* showed the highest ovicidal activity with LC$_{50}$ 10.037ppm followed by *P. purpureum* and *T. procumbens* with LC$_{50}$ values of 17.380ppm and 39.198ppm respectively [44]. The current study also investigated the pupicidal activity against the pupae of the three mosquito species selected for the study. The pupae of *An. stephensi* were more susceptible followed by *Ae. aegypti* and *Cx.quinquefasciatus* (Table 4, Fig 11). The pupicidal activity showed LC$_{50}$ and LC$_{90}$ at 29.75ppm and 65.45ppm, 53.94ppm and 97.37ppm, 18.49ppm and 58.41 ppm against the pupae of *Ae. aegypti*, *Cx. quinquefasciatus* and *An. Stephensi*, respectively (Table 5). These results are in concordance with the larvicidal and pupicidal activities exhibited by plant ethanolic extracts of *Leucas aspera* and *Bsp* against *Anopheles stephensi*. The LC$_{50}$ values were recorded as 12.732% and 0.073% for *Leucas aspera* and *Bsp* respectively [24]. The AgNPs also showed a good larvicidal activity against the larvae of three mosquito species. The percentage mortality was higher for *A. aegypti* and *A. stephensi* when compared to *C. quinquefasciatus* (Table 6, Figs 12 and 13). The LC$_{50}$ and LC$_{90}$ values were found to be 23.87ppm and 61.41ppm, 50.46ppm and 92.92 ppm, 14.98 ppm and 55.90 ppm against the 3$^{rd}$ instar larvae of *Ae. aegypti*, *Cx. quinquefasciatus* and *An. Stephensi*, respectively. LC$_{50}$ and LC$_{90}$ values recorded against the 4$^{th}$ instar larvae were 28.47ppm and 73.79ppm, 52.54ppm and 95.24ppm, 19.93ppm and 60.07ppm for *Ae. aegypti*, *Cx.*

**Table 2. Ovicidal activity of AgNPs synthesized by *Bacillus marisflavi* against the eggs of *Ae. aegypti*, *Cx. quinquefasciatus* and *An. stephensi*.**

| Conc. (ppm) | % mortality for the eggs of *Ae. aegypti* [M (SD)]* | % mortality for the eggs of *Cx quinquefasciatus* [(M (SD)]* | % mortality for the eggs of *An. stephensi* [(M (SD)]* |
|---|---|---|---|
| 5 | 0(0.00) | 3(3.82) | 0(0.00) |
| 10 | 0(0.00) | 11(2.00) | 2(2.31) |
| 20 | 13(3.82) | 18(2.31) | 12(3.26) |
| 30 | 19(2.00) | 27(3.82) | 16(3.26) |
| 40 | 24(3.26) | 35(3.82) | 24(4.61) |
| 50 | 37(3.82) | 48(3.26) | 34(2.31) |
| 60 | 41(2.00) | 58(2.31) | 63(3.82) |
| 70 | 51(3.82) | 69(5.03) | 76(3.26) |
| 80 | 67(2.00) | 76(3.26) | 78(2.31) |

* Mean (Standard Deviation).

## % mortality of 3rd instar larvae

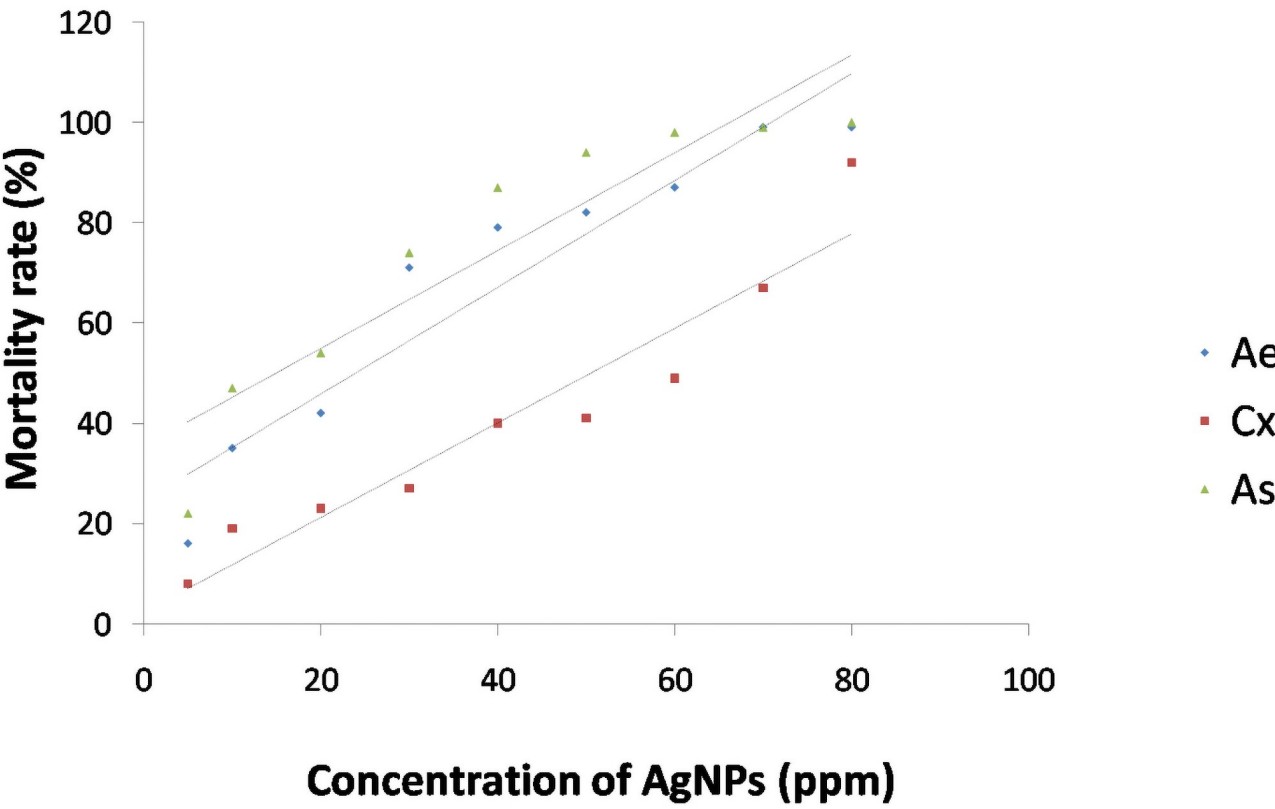

**Fig 10. Mortality curves for ovicidal activity of AgNPs synthesized by *Bacillus marisflavi* against the eggs of *Ae. aegypti*, *Cx. quinquefasciatus* and *An. Stephensi*.**

*quinquefasciatus* and *An. Stephensi*, respectively (Table 7). The AgNPs synthesized with the aqueous extracts of *Moringa oleifera* showed excellent larvicidal activity against the 3rd and 4th instar larvae of *An.gambiae*. The $LC_{50}$ and $LC_{95}$ values were 0.39ppm and 0.62 ppm. The percentage of mortality was recorded between 88%-100% when exposed for 24hrs [45]. Similar results were also obtained for the leaf extracts of *Plumbago auriculata* [46], *Zingiber officinale*, *Syzygium aromaticum* and *Datura stramonium* [47], *Blumea mollis* [48] and *Rhazya stricta*

**Table 3. Lethal concentrations, $R^2$, Regression equations and χ2 values for Ovicidal activity of AgNPs synthesized by *Bacillus marisflavi* against *Ae. aegypti*, *Cx. quinquefasciatus* and *An. stephensi*.**

| Mosquito species | $LC_{50}$ (LCL-UCL)* | $LC_{90}$ (LCL-UCL)* | $R^2$ | Regression equation | χ2 (df = 8) |
|---|---|---|---|---|---|
| *Ae. aegypti* | 13.96 (5.23–0.44) | 63.31 (57.25–1.31) | 0.951 | y = 0.810x+38.68 | 33.13 |
| *Cx. quinquefasciatus* | 24.54 (19.75–28.63) | 69.93 (64.80–6.38) | 0.973 | y = 0.881x+28.37 | 22.48 |
| *An. stephensi* | 29.14 (20.57–35.94) | 65.84 (57.72–7.86) | 0.917 | y = 1.090x+18.24 | 18.00 |

Note: $LC_{50}$- lethal concentration that kills 50% of the exposed larvae; $LC_{90}$- lethal concentration that kills 90% of the exposed larvae; LCL–Lower confidential limit; UCL–Upper confidential limit;

*—95% Confidence interval; χ2- Chi-square; df- Degrees of freedom; Table value at 0.05%–15.507.

**Table 4. Pupicidal Activity of AgNPs synthesized by *Bacillus marisflavi* against the pupae of *Ae. aegypti*, *Cx. quinquefasciatus and An. stephensi*.**

| Conc. (ppm) | % mortality for the pupae of *Ae.aegypti* [M (SD)]* | % mortality for the pupae of *Cx.quinquefasciatus* [M (SD)]* | % mortality for the pupae of *An.stephensi* [M (SD)]* |
|---|---|---|---|
| 5 | 10(2.31) | 6(2.31) | 19(3.82) |
| 10 | 29(2.00) | 15(2.00) | 42(2.31) |
| 20 | 36(3.26) | 19(2.00) | 52(3.26) |
| 30 | 60(3.26) | 25(3.82) | 72(3.26) |
| 40 | 71(3.82) | 37(3.82) | 83(2.00) |
| 50 | 78(2.3) | 41(2.00) | 90(2.31) |
| 60 | 85(3.82) | 48(3.26) | 94(2.31) |
| 70 | 93(2.00) | 63(3.82) | 98(2.31) |
| 80 | 97(2.00) | 85(3.82) | 99(2.00) |

* Mean(Standard Deviation).

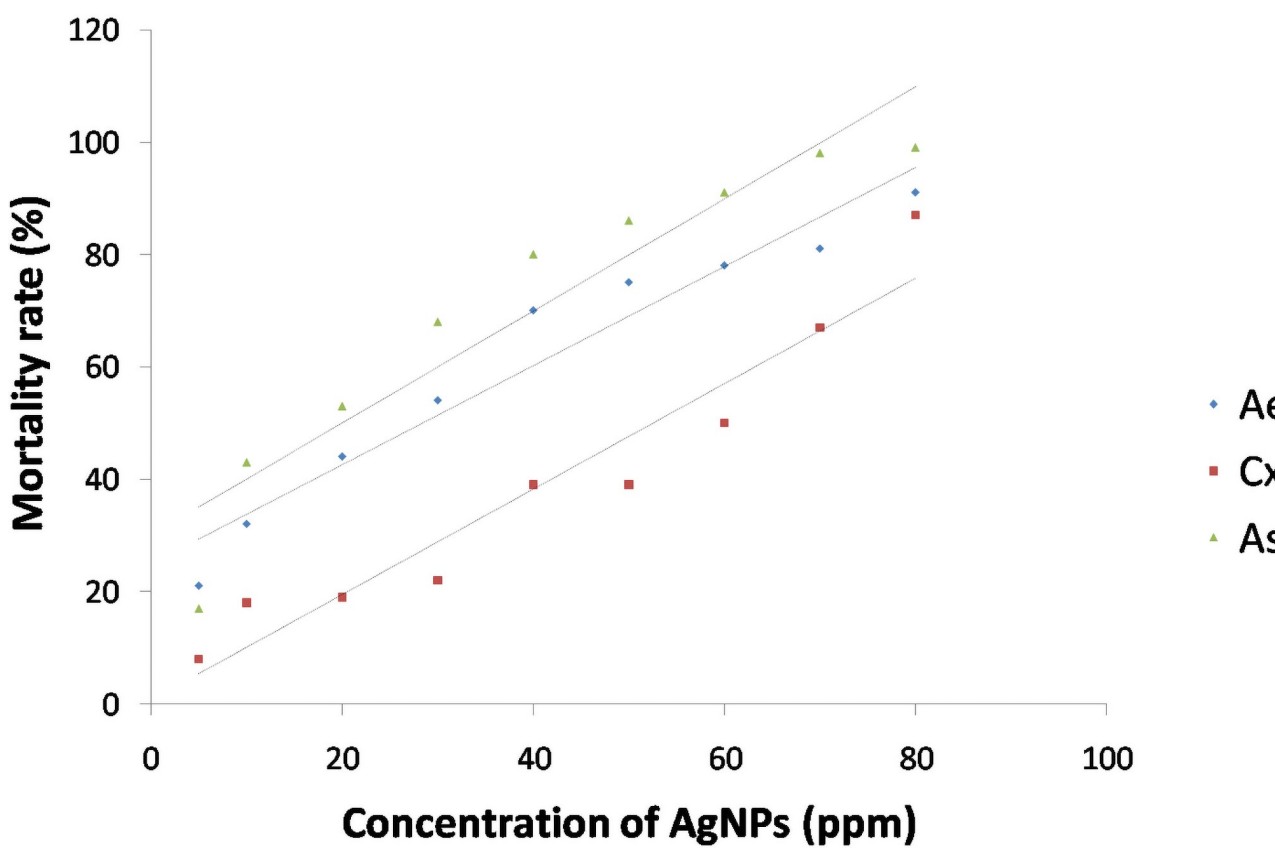

**Fig 11. Mortality curves for pupicidal activity of AgNPs synthesized by *Bacillus marisflavi* against the pupae of *Ae. aegypti*, *Cx. quinquefasciatus* and *An. Stephensi*.**

**Table 5. Lethal concentrations, $R^2$, Regression equations and χ2 values for pupicidal activity of AgNPs synthesized by *Bacillus marisflavi* against the pupae of *Ae. aegypti*, *Cx. quinquefasciatus* and *An. stephensi*.**

| Mosquito species | LC$_{50}$ (LCL-UCL)* | LC$_{90}$ (LCL-UCL)* | $R^2$ | Regression equation | χ2 (df = 8) |
|---|---|---|---|---|---|
| *Ae. Aegypti* | 29.75 (22.69–35.60) | 65.45 (58.39–75.28) | 0.938 | y = 1.120x+16.67 | 19.13 |
| *Cx. quinquefasciatus* | 53.94 (48.57–60.48) | 97.37 (86.61–113.0) | 0.950 | y = 0.921x+0.289 | 20.47 |
| *An. stephensi* | 18.49 (3.48–27.90) | 58.41 (49.50–71.85) | 0.876 | y = 1.002x+31.47 | 22.22 |

Note: LC$_{50}$- lethal concentration that kills 50% of the exposed larvae; LC$_{90}$- lethal concentration that kills 90% of the exposed larvae; LCL–Lower confidential limit; UCL–Upper confidential limit;

*—95% Confidence interval; χ2- Chi-square; df- Degrees of freedom; Table value at 0.05%–15.507.

[49]. Among the microbial AgNPs studied, the AgNPs synthesized by *Bacillus thuringiensis* and *Beauveria bassiana* had larvicidal activity against *Ae. aegypti* [50, 51]. In the present study the ovicidal, pupicidal and larvicidal activities were compared with the AgNPs synthesized by *Bt* (Positive control) that showed a good number of mortality in all the observations (S1, S3 and S5 Tables, S1–S4 Figs). The LC$_{50}$ and LC$_{90}$ values for ovicidal activity were 36.81ppm and 75.38ppm; 27.60ppm and 79.42ppm; 30.33ppm and 67.14ppm (S2 Table); the LC$_{50}$ and LC$_{90}$ values for pupicidal activity were 18.36ppm and 56.77ppm; 16.74ppm and 57.57ppm; 17.26ppm and 57.78ppm (S4 Table); the LC$_{50}$ and LC$_{90}$ values for larvicidal activity were 13.65ppm and 54.01ppm; 13.96ppm and 56.51ppm; 13.27ppm and 55.07ppm (3$^{rd}$ instar), 18.21ppm and 56.89ppm; 19.48ppm and 57.80ppm; 22.83ppm and 59.29ppm (4$^{th}$ instar) (S6 Table) against the egg, pupae and larvae of *Ae. aegypti*, *Cx. quinquefasciatus* and *An. Stephensi*, respectively. The AgNPs synthesized by *Bt* isolated from soil samples also showed larvicidal activity against the 1$^{st}$ instar to 4$^{th}$ instar larvae of *Ae.aegypti* and *Cx. quinquefasciatus* [50]. Other bacterial species such as *Pseudomonas sp.*, *Bacillus sp.* and *Lactobacillus sp.* isolated from soil and milk showed larvicidal activity against the larvae of *Ae. aegypti*. The LC$_{50}$ and LC$_{90}$ values of the AgNPs from *Lactobacillus sp.* were 8.812 μl/ml and 38.066 μl/ml, from *Bacillus sp.* were 0.343 μl/ml and 1.169 μl/ml and from *Pseudomonas sp.* were 0.473 μl/ml and 1.266 μl/ml [52]. The significance of the study was statistically tested using the Graphpad Prism 8 software.

**Table 6. Larvicidal Activity of AgNPs synthesized by *Bacillus marisflavi* against *Ae. aegypti*, *Cx. quinquefasciatus* and *An. stephensi*.**

| Conc.(ppm) | % mortality of the larval instars of *Ae. Aegypti* [M(SD)]* | | % mortality of the larval instars of *Cx. quinquefasciatus* [M(SD)]* | | % mortality of the larval instars of *An. stephensi* [M(SD)]* | |
|---|---|---|---|---|---|---|
| | Third instar | Fourth instar | Third instar | Fourth instar | Third instar | Fourth instar |
| 5 | 16(3.26) | 21(2.00) | 8(3.26) | 8(3.26) | 22(4.00) | 17(2.00) |
| 10 | 35(3.82) | 32(3.26) | 19(6.00) | 18(5.16) | 47(5.03) | 43(5.03) |
| 20 | 42(5.16) | 44(3.26) | 23(6.00) | 19(6.00) | 54(6.92) | 53(3.82) |
| 30 | 71(3.82) | 54(2.30) | 27(6.00) | 22(2.30) | 74(6.92) | 68(5.65) |
| 40 | 79(3.82) | 70(2.30) | 40(6.53) | 39(8.24) | 87(6.00) | 80(3.26) |
| 50 | 82(5.16) | 75(2.00) | 41(3.83) | 39(5.03) | 94(2.30) | 86(2.30) |
| 60 | 87(6.00) | 78(2.30) | 49(3.82) | 50(2.30) | 98(2.30) | 91(3.82) |
| 70 | 99(2.00) | 81(3.82) | 67(3.82) | 67(6.00) | 99(2.00) | 98(2.30) |
| 80 | 100(0.00) | 91(2.00) | 92(3.26) | 87(5.03) | 100(0.00) | 99(2.00) |

* Mean (Standard Deviation).

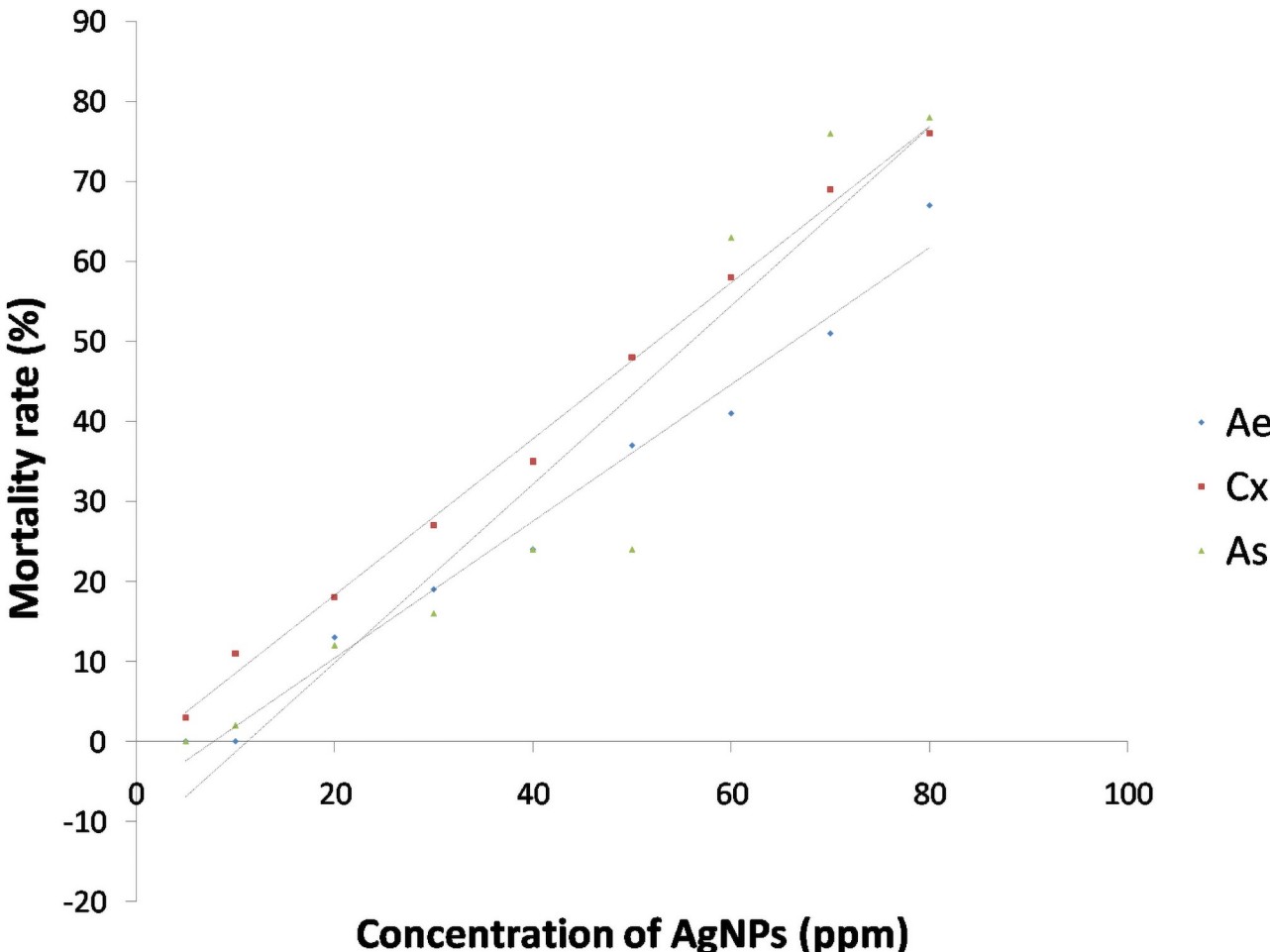

**Fig 12. Mortality curves for larvicidal activity of AgNPs synthesized by *Bacillus marisflavi* against 3$^{rd}$ instar larvae of *Ae. aegypti*, *Cx. quinquefasciatus* and *An. Stephensi*.**

The chi square analysis of $LC_{50}$ and $LC_{90}$ showed P<0.05 for all observations indicating highly significant results. This shows that observed and the predicted values of response are correlated to a good extent. Therefore, the present study revealed that the nanoparticles have an excellent toxic effect against the human vector mosquitoes. Through this study, it is evident that this rapid synthesis of nanoparticles would be an appropriate eco-friendly tool for biocontrol of mosquitoes. Formulations prepared by this method can be used as a substitute for synthetic chemicals that pose harmful effects to the environment.

## Conclusion

Mosquitoes are vectors for the majority of diseases that are considered to be a threat to human life. While defeating mosquitoes is the ultimate goal in many tropical regions, the application of different methods in achieving this remains largely impractical. Controlling the immature stages of the mosquito species will reduce the incidence and spread of these diseases. An ideal insecticide must possess toxic activity against the target organism as well as it must be inexpensive and less toxic to the environment. In the present study, the results proved that the AgNPs synthesized by biogenic mode has both the properties of a good insecticide. The use of AgNPs

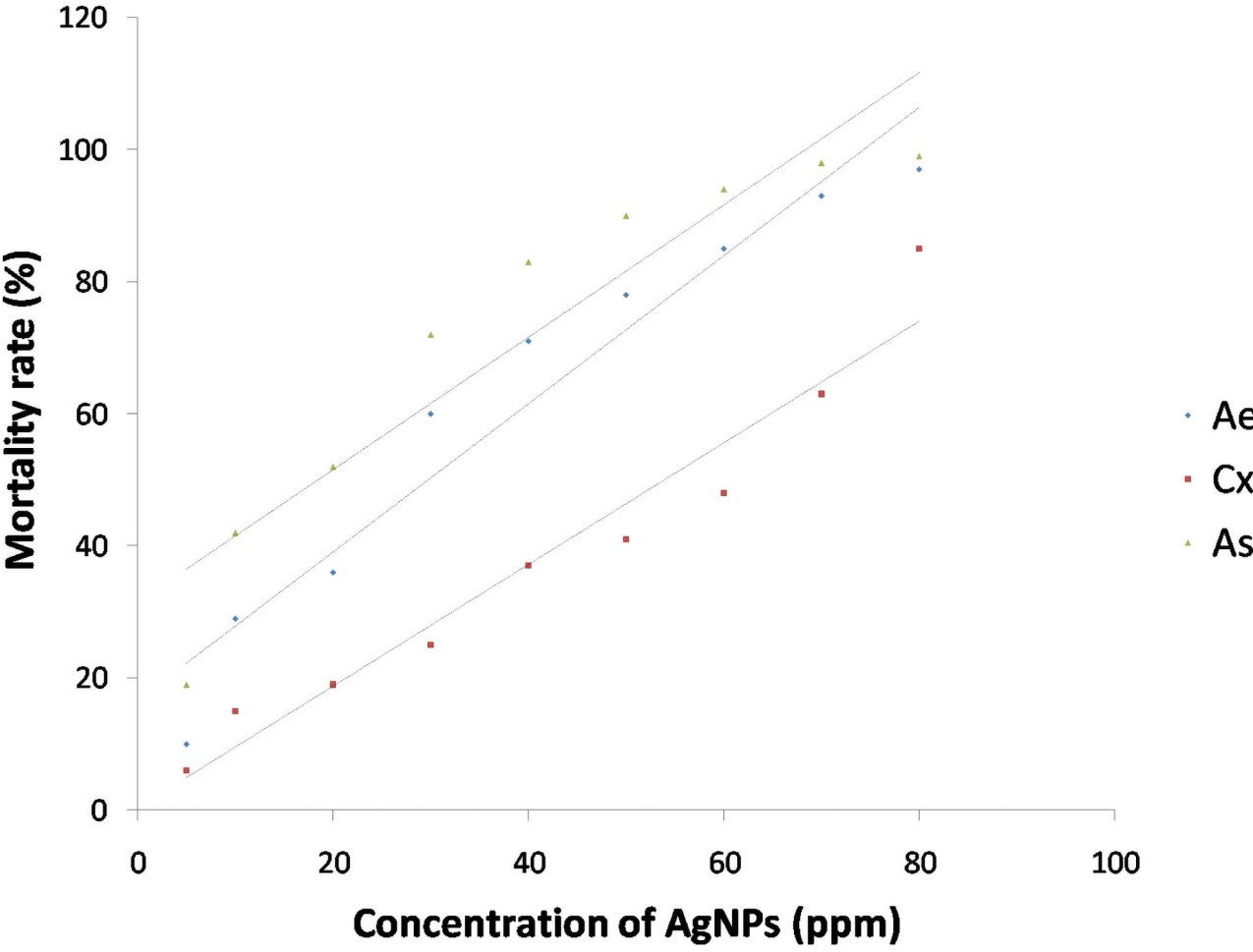

**Fig 13. Mortality curves for larvicidal activity of AgNPs synthesized by *Bacillus marisflavi* against 4$^{th}$ instar larvae of *Ae. aegypti*, *Cx. quinquefasciatus* and *An. Stephensi*.**

from marine *Bacillus* shown in this work has probably some advantages compared to other methods such as introduction of exotic species to eradicate the mosquitoes. However, the current research has some limitations. The cost-effective analysis and possible contamination of nanoparticles in the ambient or their effects on other useful microbes could be discussed

**Table 7. Lethal concentrations, R$^2$, Regression equations and χ2 values for larvicidal activity of AgNPs *synthesized* by *Bacillus marisflavi* against *Ae. aegypti*, *Cx. quinquefasciatus* and *An. stephensi*.**

| Larval species | Larval instars | LC$_{50}$ (LCL-UCL)* | LC$_{90}$ (LCL-UCL)* | R$^2$ | Regression equation | χ2 (df = 8) |
|---|---|---|---|---|---|---|
| *Ae. aegypti* | 3$^{rd}$ | 23.87 (12.89–31.69) | 61.41 (53.16–73.62) | 0.902 | y = 1.066x+24.56 | 16.04 |
| | 4$^{th}$ | 28.47 (21.56–34.15) | 73.79 (65.95–85.02) | 0.942 | y = 0.882x+24.87 | 19.05 |
| *Cx.quinquefasciatus* | 3$^{rd}$ | 50.46 (43.74–58.70) | 92.92 (80.38–113.5) | 0.918 | y = 0.942x+2.459 | 16.75 |
| | 4$^{th}$ | 52.54 (46.28–60.30) | 95.24 (83.19–114.2) | 0.930 | y = 0.936x+0.791 | 16.51 |
| *An. stephensi* | 3$^{rd}$ | 14.98 (3.57–25.63) | 55.90 (46.50–70.04) | 0.856 | y = 0.977x+35.36 | 26.04 |
| | 4$^{th}$ | 19.93 (6.98–28.5) | 60.07 (51.62–72.65) | 0.893 | y = .996x+30.14 | 21.30 |

LC$_{50}$- lethal concentration that kills 50% of the exposed larvae; LC$_{90}$- lethal concentration that kills 90% of the exposed larvae; LCL–Lower confidential limit; UCL–Upper confidential limit;

*—95% Confidence interval; χ2- Chi-square; df- Degrees of freedom; Table value at 0.05%–15.507.

briefly. In conclusion, the present study confirms that the silver nanoparticles synthesized by *Bacillus marisflavi* prove to be an effective eco-friendly tool in controlling human vector mosquitoes.

## Supporting information

**S1 Table. Ovicidal activity of AgNPs synthesized by *Bacillus thuringiensis* against the eggs of *Ae. aegypti, Cx. quinquefasciatus* and *An. stephensi*.**
(DOCX)

**S2 Table. Lethal concentrations, $R^2$, Regression equations and χ2 values for Ovicidal activity of AgNPs synthesized by *Bacillus thuringiensis* against *Ae. aegypti, Cx. quinquefasciatus and An. stephensi*.**
(DOCX)

**S3 Table. Pupicidal Activity of AgNPs synthesized by *Bacillus thuringiensis* against the pupae of *Ae. aegypti, Cx. quinquefasciatus* and *An. stephensi*.**
(DOCX)

**S4 Table. Lethal concentrations, $R^2$, Regression equations and χ2 values for pupicidal activity of AgNPs synthesized by *Bacillus thuringiensis* against the pupae of *Ae. aegypti, Cx. quinquefasciatus and An. stephensi*.**
(DOCX)

**S5 Table. Larvicidal Activity of AgNPs synthesized by *Bacillus thuringiensis* against *Ae. aegypti, Cx. quinquefasciatus and An. stephensi*.**
(DOCX)

**S6 Table. Lethal concentrations, $R^2$, Regression equations and χ2 values for larvicidal activity of AgNPs *synthesized* by *Bacillus thuringiensis* against *Ae. aegypti, Cx. quinquefasciatus and An. stephensi*.**
(DOCX)

**S1 Fig. Mortality curves for ovicidal activity of AgNPs synthesized by *Bacillus thuringiensis* against *Ae. aegypti, Cx. quinquefasciatus and An. stephensi*.**
(PDF)

**S2 Fig. Mortality curves for pupicidal activity of AgNPs synthesized by *Bacillus thuringiensis* against the pupae of *Ae. aegypti, Cx. quinquefasciatus* and *An. stephensi*.**
(PDF)

**S3 Fig. Mortality curves for larvicidal activity of AgNPs synthesized by *Bacillus thuringiensis* against 3rd instar larvae of *Ae. aegypti, Cx. quinquefasciatus and An. stephensi*.**
(PDF)

**S4 Fig. Mortality curves for larvicidal activity of AgNPs synthesized by *Bacillus thuringiensis* against 4th instar larvae of *Ae. aegypti, Cx. quinquefasciatus and An. stephensi*.**
(PDF)

## Acknowledgments

Authors thank the Management of Thiagarajar College, Madurai for providing all necessary facilities to carry out the research successfully.

## Author Contributions

**Conceptualization:** Thelma J.

**Data curation:** Thelma J.

**Formal analysis:** Thelma J.

**Investigation:** Thelma J.

**Methodology:** Thelma J.

**Project administration:** Balasubramanian C.

**Supervision:** Balasubramanian C.

**Validation:** Balasubramanian C.

**Writing – original draft:** Thelma J.

**Writing – review & editing:** Balasubramanian C.

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
