## [Decision Letter · Decision Letter 0]

22 Mar 2021

PONE-D-21-03066

Ovicidal, larvicidal and pupicidal efficacy of silver nanoparticles synthesized by Bacillus marisflavi against the chosen mosquito species

PLOS ONE

Dear Dr. J,

Thank you for submitting your manuscript to PLOS ONE. After careful consideration, we feel that it has merit but does not fully meet PLOS ONE’s publication criteria as it currently stands. Therefore, we invite you to submit a revised version of the manuscript that addresses the points raised during the review process.

We look forward to receiving your revised manuscript.

Kind regards,

Jiang-Shiou Hwang, Ph.D.

Academic Editor

PLOS ONE

Journal Requirements:

2. In your Methods section, please provide additional information regarding the permits you obtained for the work. Please ensure you have included the full name of the authority that approved the sampling site access and, if no permits were required, a brief statement explaining why.

3. We note you have included a table to which you do not refer in the text of your manuscript. Please ensure that you refer to Tables in your text; if accepted, production will need this reference to link the reader to the Table.

Additional Editor Comments (if provided):

Reviewers' comments:

Reviewer's Responses to Questions

**Comments to the Author**

1. Is the manuscript technically sound, and do the data support the conclusions?

Reviewer #1: Yes

Reviewer #2: Yes

2. Has the statistical analysis been performed appropriately and rigorously? 

Reviewer #1: Yes

Reviewer #2: No

3. Have the authors made all data underlying the findings in their manuscript fully available?

Reviewer #1: Yes

Reviewer #2: Yes

4. Is the manuscript presented in an intelligible fashion and written in standard English?

Reviewer #1: Yes

Reviewer #2: No

5. Review Comments to the Author

Reviewer #1: The study is well performed and the meaning and the logic are clear. The authors prepared the Ag nanoparticles using Bacillus spp is very interesting. However, I am providing some minor suggestions to improve the paper below:

Correction needed:

AFM results need some revisions.

The Conclusion need to be rewritten

Noticed, numerous typographical grammatical errors. Authors please revise the whole stuff.

Ensure italics of organisms genus and species name throughout the manuscript.

Check for references as per the journal format.

Line no. 425-27. Please check the reference number 17

Authors, please add recent references

Reviewer #2: The present manuscript explains that the nanoparticles have an excellent toxic effect against the vector mosquitoes. After synthesis of silver nano particle from Bacillus so the toxicity evaluation has been doen against egg, larvae and pupae of disease causing mosquito species. The present study advocates that the rapid synthesis of AgNPs would be an

appropriate eco-friendly tool for biocontrol of vector mosquitoes.

However after careful perusal of the manuscript I have following observations:

Abstract line 11: disease to humans, better to say disease in humans

Line 20-22 : Sentence not complete: LC50 and LC90 values for the ovicidal, larvicidal and pupicidal efficacy of the AgNPs against the egg, larvae and pupae of Aedes aegypti, Culex 2 quinquefasciatus and Anopheles stephensi respectively.

Line 21-22 : All the observations were statistically 23 significant at P<0.05. Is it all the observations or particular result :

Line 24-26 : Hence, it is evident that this the rapid synthesis of AgNPs would be an appropriate eco-friendly tool for biocontrol of vector mosquitoes .

Line 34-37: Mosquitoes of genus Anopheles (An.) serve as a suitable host for Plasmodium parasite and are 35 responsible for transmitting Malaria. Culex (Cx.) is another genus of mosquito that acts as a host 36 for Wuchereria bancrofti, which transmits Lymphatic filariasis and Aedes (Ae.) that harbors the 37 virus belonging to flaviviridae family and are responsible for causing Dengue.

Line 40: chemical methods; but these these

methods aggravate the resistance by in mosquitoes,

Line 66-67: Therefore, the present study is intended to explore the nanoparticle synthesizing ability of marine Bacillus species and their efficacy is evaluated against vectorvarious stages such as egg, larva and pupa of Ae. aegypti, Cx. quinquefasciatus and An. stephensi 69 (L.) respectively under laboratory condition.

Method section is missing

Results and discussion:

Line 73-74 : marine seawater

Method, results and discussions are mixed

The significance of the regression coefficients can be determined by the P 176 value. P-value is defined as the smallest level of significance leading to rejection of null 177 hypothesis. A small P value indicates a significant study [53]. All the results showed P

Line 175-180: The significance of the regression coefficients can be determined by the P valueP value. P-value is defined as the smallest level of significance leading to rejection of null hypothesisnull hypothesis. A small P value indicates a significant study [53]. All the results showed P

Table 2-7: should be presented as probit regression graph (mortality vs concentration) only These tables just present the raw data , further the value of SD should have n value in the parentheses

Materials and methods presented after results and discussion is very confusing and typo errors are seen , linguistic information is required

6. PLOS authors have the option to publish the peer review history of their article (what does this mean?). If published, this will include your full peer review and any attached files.

Reviewer #1: No

Reviewer #2: No

---

## [Author Response · Author response to Decision Letter 0]

21 May 2021

The manuscript has been revised according to the comments given by the reviewers.

---

## [Decision Letter · Decision Letter 1]

12 Oct 2021

PONE-D-21-03066R1Ovicidal, larvicidal and pupicidal efficacy of silver nanoparticles synthesized by Bacillus marisflavi against the chosen mosquito speciesPLOS ONE

Dear Dr. J,

Thank you for submitting your manuscript to PLOS ONE. After careful consideration, we feel that it has merit but does not fully meet PLOS ONE’s publication criteria as it currently stands. Therefore, we invite you to submit a revised version of the manuscript that addresses the points raised during the review process. Please submit your revised manuscript by Nov 26 2021 11:59PM. If you will need more time than this to complete your revisions, please reply to this message or contact the journal office at plosone@plos.org. Please include the following items when submitting your revised manuscript:A rebuttal letter that responds to each point raised by the academic editor and reviewer(s). You should upload this letter as a separate file labeled 'Response to Reviewers'.A marked-up copy of your manuscript that highlights changes made to the original version. You should upload this as a separate file labeled 'Revised Manuscript with Track Changes'.An unmarked version of your revised paper without tracked changes. You should upload this as a separate file labeled 'Manuscript'.If applicable, we recommend that you deposit your laboratory protocols in protocols.io to enhance the reproducibility of your results. Protocols.io assigns your protocol its own identifier (DOI) so that it can be cited independently in the future. For instructions see: https://journals.plos.org/plosone/s/submission-guidelines#loc-laboratory-protocols. Additionally, PLOS ONE offers an option for publishing peer-reviewed Lab Protocol articles, which describe protocols hosted on protocols.io. Read more information on sharing protocols at https://plos.org/protocols?utm_medium=editorial-email&utm_source=authorletters&utm_campaign=protocols.

We look forward to receiving your revised manuscript.

Kind regards,

Jiang-Shiou Hwang, Ph.D.

Academic Editor

PLOS ONE

Journal Requirements:

Reviewers' comments:

Reviewer's Responses to Questions

**Comments to the Author**

1. If the authors have adequately addressed your comments raised in a previous round of review and you feel that this manuscript is now acceptable for publication, you may indicate that here to bypass the “Comments to the Author” section, enter your conflict of interest statement in the “Confidential to Editor” section, and submit your "Accept" recommendation.

Reviewer #1: All comments have been addressed

Reviewer #3: All comments have been addressed

2. Is the manuscript technically sound, and do the data support the conclusions?

Reviewer #1: Yes

Reviewer #3: Yes

3. Has the statistical analysis been performed appropriately and rigorously? 

Reviewer #1: Yes

Reviewer #3: Yes

4. Have the authors made all data underlying the findings in their manuscript fully available?

Reviewer #1: Yes

Reviewer #3: Yes

5. Is the manuscript presented in an intelligible fashion and written in standard English?

Reviewer #1: Yes

Reviewer #3: Yes

6. Review Comments to the Author

Reviewer #1: The authors clearly addressed to potential reviewer's comment. Now, this manuscript is precise and can be considered for publication in PLOS ONE.

Reviewer #3: I do not remember having reviewed this manuscript previously. In any case, I have gone through the manuscript that has already been reviewed by two others. In the R1 version of the manuscript, the authors have carefully taken into account of all the suggestions made by two reviewers. This has improved considerably the readability of this contribution.

I have a few suggestions that the authors may have to consider about the application of the present work to the field conditions. The limitations suggested here may be added to the conclusion unit or the discussion section.

1. While defeating mosquitoes is the ultimate goal in many tropical regions, the application of different methods in achieving this remains largely impractical.

2. The use of silver nanoparticles from Bacillus shown in this work has probably some advantages compared to other methods such as introduction of exotic fish species to eradicate the mosquitoes

3. However, authors may have to provide cost-effective analysis and possible contamination of nanoparticles in the ambient or their effects on other useful microbes.

7. PLOS authors have the option to publish the peer review history of their article (what does this mean?). If published, this will include your full peer review and any attached files.

Reviewer #1: No

Reviewer #3: No

---

## [Author Response · Author response to Decision Letter 1]

29 Oct 2021

I have modified the conclusion according to the suggestions given by Reviewer 3

---

## [Editor Report · Decision Letter 2]

8 Nov 2021

Ovicidal, larvicidal and pupicidal efficacy of silver nanoparticles synthesized by Bacillus marisflavi against the chosen mosquito species

PONE-D-21-03066R2

Dear Dr. J,

We’re pleased to inform you that your manuscript has been judged scientifically suitable for publication and will be formally accepted for publication once it meets all outstanding technical requirements.

Kind regards,

Jiang-Shiou Hwang, Ph.D.

Academic Editor

PLOS ONE
---

## [Editor Report · Acceptance letter]

23 Nov 2021

PONE-D-21-03066R2 

Ovicidal, larvicidal and pupicidal efficacy of silver nanoparticles synthesized by *Bacillus marisflavi* against the chosen mosquito species 

Dear Dr. J:

I'm pleased to inform you that your manuscript has been deemed suitable for publication in PLOS ONE. Congratulations! Your manuscript is now with our production department. 

Kind regards, 

on behalf of

Prof. Jiang-Shiou Hwang 

Academic Editor

PLOS ONE